# The Importance of Energy Theory in Shaping Elevational Species Richness Patterns in Plants

**DOI:** 10.3390/biology11060819

**Published:** 2022-05-26

**Authors:** Zihan Jiang, Qiuyu Liu, Wei Xu, Changhui Peng

**Affiliations:** 1Department of Biology Sciences, Institute of Environment Sciences, University of Quebec at Montreal, C.P. 8888, Succ. Centre-Ville, Montreal, QC H3C 3P8, Canada; jzh1983427@gmail.com (Z.J.); qiuyuliu0711@gmail.com (Q.L.); 2Guangzhou Institute of Geography, Guangdong Academy of Sciences, Guangzhou 510075, China; xuwei_1012@163.com; 3School of Geographic Sciences, Hunan Normal University, Changsha 410081, China

**Keywords:** elevational gradient, multi-taxa, plant, global, energy availability

## Abstract

**Simple Summary:**

Mountains are storehouses of global biodiversity; they host one-quarter of all terrestrial species at condensed spatial scales. Understanding the mechanisms and patterns that are associated with species richness along elevational gradients would help us to answer why mountains are so biologically diverse. However, despite decades of effort, a consensus regarding the processes that contribute to diversity in these locations is yet to be reached. Here, we compare the importance of four hypotheses on shaping the elevational species richness patterns of plants in 22 global mountainous regions. Moreover, a comparative analysis among different plant growth form groups was conducted to remove the influence of specific mountain attributes, such as topography complexity, history, and area. The results reveal that the performance of these hypotheses depend on the growth form and climatic conditions. The energy hypothesis provides a better explanation for the elevational species richness patterns of woody plants than the other hypotheses, while energy-related factors show greater explanatory power for predicting elevational species richness patterns in colder mountain regions. Our results highlight the importance of energy availability for the production of plant elevational species richness patterns, which is essential for the conservation and management of biodiversity.

**Abstract:**

Many hypotheses have been proposed to explain elevational species richness patterns; however, evaluating their importance remains a challenge, as mountains that are nested within different biogeographic regions have different environmental attributes. Here, we conducted a comparative study for trees, shrubs, herbs, and ferns along the same elevational gradient for 22 mountains worldwide, examining the performance of hypotheses of energy, tolerance, climatic variability, and spatial area to explain the elevational species richness patterns for each plant group. Results show that for trees and shrubs, energy-related factors exhibit greater explanatory power than other factors, whereas the factors that are associated with climatic variability performed better in explaining the elevational species richness patterns of herbs and ferns. For colder mountains, energy-related factors emerged as the main drivers of woody species diversity, whereas in hotter and wetter ecosystems, temperature and precipitation were the most important predictors of species richness along elevational gradients. For herbs and ferns, the variation in species richness was less than that of woody species. These findings provide important evidence concerning the generality of the energy theory for explaining the elevational species richness pattern of plants, highlighting that the underlying mechanisms may change among different growth form groups and regions within which mountains are nested.

## 1. Introduction

How do species richness patterns vary along elevational gradients? What are the underlying reasons for elevational species richness patterns? These questions have fascinated many ecologists and biogeographers over the last few decades [1,2,3,4,5]; however, the mechanism responsible remains elusive, partly because species richness with elevation often exhibits more complex patterns [6,7]. The latitudinal gradient of species richness does not always decrease linearly, and the relationship between elevation and species richness differs among mountains and taxa. This relationship can be hump-shaped, decreasing linearly, flat-horizontal then decreasing, and increasing, which reduces the likelihood of any consensus on the underlying reason for the elevational species richness pattern [8].

Many factors have been recognized as being the main drivers of elevational species richness patterns; however, these can be classified into four basics categories. Area [3,9]: this hypothesis suggests that the variation in species richness results from the shape of the mountain—the greater the area, the greater the number of species that can be supported. Climatic variability [10,11]: this hypothesis suggests that habitats with greater climatic variation may benefit from niche partitioning, thereby resulting in a more diverse range of species coexisting. Tolerance [12,13]: this hypothesis suggests that habitats with harsh environments would remove intolerant species, whereas more species are allowed to coexist in a benign environment. Energy [14]: this hypothesis suggests that a habitat with more energy availability would support the survival of a greater number of individuals, which in turn supports more species in the area. Among these hypotheses, the energy theory has been widely accepted [15,16,17], since for any organism, energy intake involves many resource requirements. For example, plant species require adequate temperatures and the absorption of water and soil nutrients to fix the energy stored in photons. These resources may vary among mountains, and they can become limiting factors in shaping the altitudinal patterns of plant species richness.

Mountain gradients differ globally by geological age, elevational range, and disturbance history, as well as topographic and environmental attributes [7]. Moreover, the environmental gradient with elevation varies among mountains in different climatic zones. For example, the precipitation along an elevational gradient usually exhibits a hump-shaped pattern in an arid ecosystem, but this can also decrease with elevation in tropical zones [18]. It is possible that elevational species richness patterns among mountains may be shaped by the same mechanism, and various patterns can result from differences in elevational gradients.

A promising approach to better understand the factors determining elevational species richness patterns is a comparative study of contrasting growth form groups along the same elevational transect [6,19], thus minimizing the influence of different regional factors at different elevational transects. In addition, ecological requirements and responses to environmental changes differ among growth form groups; for example, herb species such as orchids prefer a shaded habitat, whereas tree species require high levels of light availability. Therefore, analyzing elevational species richness patterns for various growth form groups may be helpful in evaluating the generality of species richness hypotheses.

Here, we conducted a comparative analysis of elevational species richness patterns for plants with different growth forms (trees, shrubs, ferns, and herbs) along the same elevational transect. These taxa comprise a range of body sizes, dispersal abilities, and habitat types, and they differ in their ecological requirements and responses to environmental change. Such a comparison among plant growth forms would help to improve our understanding of the differences in elevational species richness patterns among plant growth form groups. Furthermore, we estimated the importance of the energy hypothesis in explaining the elevational species richness pattern of plants, by comparing the performance of the other three hypotheses, which have also been widely considered as drivers of elevational species richness patterns. We asked three questions: (1) how does taxon richness vary with elevation? (2) Are the elevational patterns of species richness for different taxa shaped via different mechanisms? (3) Which hypotheses have better generality in explaining the variation in plant species richness with elevation?

## 2. Materials and Methods

### 2.1. Data Collection

A literature survey for target studies from 1970–2020 was conducted through Google Scholar, ISI Web of Science, and the China National Knowledge Infrastructure (http://www.cnki.net (8 June 2020)); (elevatio* or altitud*), (richness or diversit*), and (gradien* or patter* or transec* or varian*) were used as keywords. A total of 10,020 studies were identified. We further selected studies based on the following criteria: (1) the study described plant species richness patterns along a single elevational gradient; (2) the species richness data was collected in the field rather than from the literature or from museums; (3) the elevational range of the transects was more than 1000 asl.; (4) there were no sampling imbalance issues in the studies; (5) the number of sampling sites was more than 10 along the elevational transect—this criterion was specified because species richness along elevation also varies with sampling intensity [20]; declines or peaks in species richness may correspond to declines or peaks in sampling intensity); (6) the elevational sampling intervals were less than 500 asl.; and (7) the species included in the taxon did not have substantial differences in resource acquisition. We also selected studies that examined species richness patterns across multiple plant growth form groups (≥3) in the same mountain area. A cross-growth form comparison was then conducted to remove the influence of human disturbance, age, and topographic complexity. The plant species richness data were extracted using GetData. To ensure a one-to-one correspondence between the elevation and species richness, the species richness values for each elevational step were averaged for publications with multiple sampling plots at one elevation.

Twelve environmental factors were extracted or calculated in this study; these factors characterized the area, climatic variability, energy, and tolerance. Mean annual temperature (MAT), water vapor pressure (vapor), mean annual precipitation (MAP), and actual evapotranspiration (AET) were used to represent energy, as they are widely related to energy availability [21]. The AET was calculated according to Turc’s formula [22], where T is the mean annual temperature (°C) and P is the mean annual precipitation mm):AET = P/[0.9 + (P/L)^2^]^1/2^
L = 300 + 25T + 0.05T^3^

To account for climatic variability, isothermality (ISO) and precipitation seasonality (PS) were extracted and used to represent the variability of temperature and precipitation. We added factors that are related to extreme climate to describe the influence of tolerance [23], which were the minimum temperature (Min), maximum temperature (Max), precipitation of the driest quarter (PDQ), and precipitation of the wettest quarter (PWQ). The surface area of the mountain for each elevational change of 100 m (area, represented as the number of grid cells within an elevational increase of 100 m) was included in the analysis to represent the area [24].

The elevational data for each transect were extracted from the 90 m Shuttle Radar Topographic Mission (SRTM) altitude model, and environmental factors were collected from WorldClim [25] for each mountain at 30 s resolution (10 × 10 km^2^). To compile the climatic data for each elevational gradient, 12 candidate variables were extracted or calculated for each mountain region, and then their mean values were calculated for every 100 m elevation interval using ArcGIS. Finally, each factor was interpolated to match the elevational species richness data using the R package “zoo” [26].

### 2.2. Analysis

To access the relationship between the species richness of each taxon and elevation, we performed polynomial regressions (species richness as a linear, quadratic, and cubed function of elevation). The selection of the best polynomial regression was guided using the Akaike information criterion (AIC). Spatial correlation analysis was performed to examine the relationships among the focal taxa. We employed a generalized linear model (GLM) to evaluate the explanatory power of each candidate variable. Environmental and spatial factors were the candidate variables, whereas species richness was the response variable. We fitted GLMs for the response variables as quadratic or linear functions of each explanatory variable. For the candidate variables to show a significant quadratic and linear relationship with species richness, we compared these models using the AIC.

To determine the generality of the four hypotheses on predicting the elevational species richness pattern of the plant taxa, we calculated the average AIC value of each to assess the influence of the four hypotheses. Before carrying out this analysis, the candidate variables were selected if they were found to have a significant relationship with species richness. To determine how many variations in species richness along elevational gradient could be explained, we selected the factor that exhibited the best prediction power (lowest AIC) in each hypothesis group, and then multiple ordinary least squares regression (multiple OLS) models were used to evaluate the performance. Before conducting the multiple regression analysis, candidate explanatory variables were selected if they had a significant relationship with species richness. The collinearity among the candidate variables can influence multiple regression analyses. To overcome this issue, we first evaluated the correlations among all the significant candidate variables, for any variable-pair that was found to be strongly correlated (Pearson’s |r| > 0.70) [10]. The variable exhibiting the weaker (greater AIC) relationship with species richness was excluded.

## 3. Results

In total, we collected data on 66 elevational species richness patterns from 22 mountains around the world (Figure 1), including the elevational species richness gradients of 22 trees, 22 shrubs, 21 herbs, and 1 fern. The mountains are in different climatic regions, and their elevations range from 12 to 7032 m, their MAT ranges from −3.63–12.55 °C, and their MAP ranges from 242.4–1771.43 mm. The relationships between plant species richness patterns and elevations can be classified into four groups (Figure 2a and Appendix A): decreasing linearly, hump-shape, cubed, and no significant relationship with elevation. For trees and shrubs, a linear decrease is the most common elevational species richness pattern (tree: 50%; shrub: 41%), while both exhibiting a non-significant relationship with elevation in two elevational gradients. In contrast, the highest proportion for the elevational species richness patterns of herbs was a hump shaped relationship. Moreover, the elevational species richness, patterns of plant growth form groups within the same mountain were similar (Figure 1b). The average correlation between tree and shrub was significantly greater than for other taxa pairs, whereas the tree–herb relationship was the weakest.

The performance of different hypotheses for predicting the elevational species richness patterns varied among the plant groups (Figure 2c, Appendix A). The energy-related factors performed best (lowest average AIC) explaining the elevational species richness pattern for trees and shrubs (Figure 3), while tolerance-related factors emerged as being the second-best predictor. For herbs, climatic variability related factors show the lowest AIC in predicting their species richness pattern. Furthermore, energy-related factors performed best or second-best as univariate predictors of the elevational richness pattern for trees, in 19 out of 22 mountains. For the shrub species, the average AIC of energy-related factors was the lowest for 5 out of 22 mountains, and the second lowest for 9 mountains. For herb species, energy-related factors emerged as being the most important predictors of the elevational species richness pattern for only five mountains. Energy-related factors were the second most important predictor in five mountains, while no energy-related factors showed a significant relationship with the elevational species richness pattern of herbs in eight mountains.

As expected, the candidate factors in this study were highly correlated (|r| > 0.70, Appendix A), and after excluding the factor that indicated the higher AIC in each collinear pair, the multiple-predictor models showed that the percentage variation in the elevational richness patterns explained by the candidate factors varied among taxa (Appendix A). The elevational species richness pattern for the tree species exhibited a higher R^2^ than the other taxa for seven mountains, whereas shrubs had the highest R^2^ for four mountains. Herbs only exhibited a higher R^2^ in the Himalayan mountains (Nepal), and they had no significant relationship with any of the candidate factors for Mt. Miandam.

The performance of the elevational species richness pattern hypotheses also differed among the climatic regimes in which the mountains were located (Appendix A). The energy-related factors performed better than other factors as single predictors of the elevational species richness pattern of woody plants (tree and shrub) for the colder mountains (Mt. Changbai, Helan, and Dongling), with the exception of shrubs on Mt. Dongling, which were slightly higher than the tolerance-related factors. The multiple-predictor models support these results. The best multiple-predictor model for the elevational species richness pattern of woody plants in these mountains all included energy-related factors, namely shrubs in Mt. Helan, trees on Mt. Dongling, and trees and shrubs on Mt. Changbai. Energy-related factors emerged as the single best predictor of elevational species richness patterns. Temperature and precipitation variability were better in the hottest mountains (Mt. Popocatepetl, Miandam, and Jiuding) than the other factors.

## 4. Discussion

Our results provide novel evidence that the energy constraint may be a better explanation for the elevational species richness patterns of plants than the area, climatic variability, and tolerance. By using a comparative approach among different plant life-forms along some of the elevational gradients, conclusions could be more precise, given that the error resulting from the differences in topography, area, and history among mountains was removed. Our most important finding was that the performance of energy-related factors was much better in predicting the elevational species richness patterns of woody taxa, which is in line with the findings of recent studies [21,27,28]. This supports the contention that a variation in energy availability along elevational gradients may play a more critical role in regulating community assembly.

In our model, the performance of energy-related factors as a predictor of woody plant diversity increased from warmer to colder ecosystems, whereas temperature and precipitation variability were more important regulators of plant diversity in wetter and hotter mountains. This suggests that the energy constraint plays a crucial role in structuring woody communities, where temperature is a limiting factor. In contrast, the role of climatic variability supersedes the influence of energy constraints in more benign environments, given that it can facilitate temporal niche partitioning [29]. However, since this study only considered temperature and precipitation, the role of other factors that determine energy intake cannot be excluded. Soil nutrients are also very important for energy intake [30,31,32], and the water-holding capacity of soil influences the water availability of plants, which may be a limiting factor in arid zones. It is unlikely that a single factor regulates the species richness pattern along the elevational gradient, and Jiang et al. [33] showed that herb richness is determined by light availability at low and high elevations. However, such inferences are weakened by the lack of available data on site-specific growth habits for each mountain. Furthermore, targeted investigations of the relationships between species richness and soil variables across different mountain regions would be beneficial. 

Studying the differences in hypothesis performance among taxa is particularly important, given that it would simplify the development of the understanding of species richness patterns [34], which can be attributed to several underlying mechanisms. One explanation for this is that different taxa share similar environmental drivers. For example, the resource requirements, dispersal ability, and evolutionary history are similar between trees and shrubs, and their species richness responds similarly to elevation. Another possible cause for covariance of species richness between different taxa is ecological interdependence, such as for herbs and trees. In this case, an association with resources and habitat complexity between the two in turn may maintain the species richness of herbs [35,36,37].

Comparative studies along the same elevational gradient represent a promising approach for evaluating the mechanisms of elevational species richness patterns. Contrasting species richness patterns among studies and taxa might result from differences in sample extent and intensity [38], rather than the underlying mechanisms. Implementing the same sample design can control for potential bias. However, the “ecologically equivalent” [19] sampling approach would have a pronounced influence on the interpretability of the species richness patterns. Although plant size was different among the focal taxa, some candidate factors such as temperature, moisture, and productivity were measured at the same spatial scale, which was likely too coarse to detect herb variation in species richness. However, in some cases, environmental factors measured at a large scale can also exhibit great predictive performance on the elevational richness patterns of taxa with small body sizes [39,40,41]. This phenomenon could result from data collection; if occurrence data was obtained from interpolation, the species distribution would be overestimated [42], which would reduce the variation of species richness and make it easier to predict using large scale environmental factors.

Convergent species richness patterns between the two taxa may also result from scale effects [38]. The chosen scale of the sampled species richness will strongly affect the observed pattern, which has been confirmed by numerous studies [43,44,45]. Covariance among different plant growth forms is more common at smaller spatial scales. For example, the elevational species richness pattern of ants, birds, beetles, and moths exhibited a positive relationship in the Great Smoky Mountains, but the covariance among those taxa disappeared at larger spatial scales [19].

Our knowledge surrounding the plant species richness responses to elevation is growing. However, the underlying reasons that shape elevational species richness patterns are still a subject of considerable debate. Analyzing several contrasting taxa along the same elevational transect can enhance our understanding of the mechanisms responsible for various elevational richness patterns. However, determining an appropriate scale for each taxon remains a substantial challenge. Exploring species richness patterns along the elevational gradient at too coarse of a spatial scale would result in bias [38]. This challenge could be improved with further research on various taxa at multiple scales, requiring more ground observations and investigations. We hope that the observational evidence presented here will motivate others to investigate the strength and shape of species richness, in addition to energy availability relationships across differing scales in other mountain regions.

## 5. Conclusions

Our results revealed the importance of energy constraints in shaping the patterns of plant richness along elevational gradients, specifically for woody plants in mountains with lower temperatures. Considering the changes in temperature and precipitation resulting from global warming, as evident in this study, it is likely that a shift in the elevational gradient of energy availability would reshape the elevational species richness patterns of plants. Therefore, our ability to predict elevational richness patterns will benefit from further studies that include more energy-related factors, on explaining the elevational species richness patterns of different taxa and mountains.

## Figures and Tables

**Figure 1 biology-11-00819-f001:**
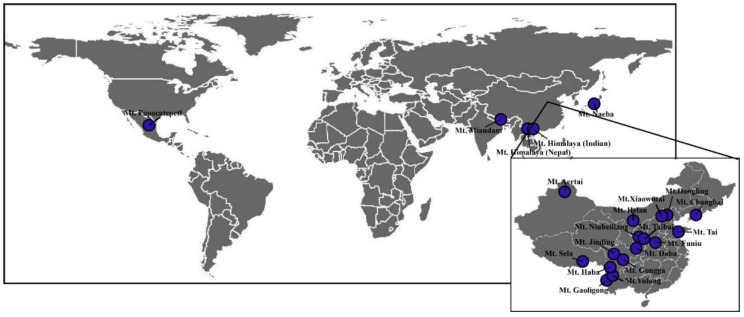
Map with locations of the 22 mountain areas included in this study.

**Figure 2 biology-11-00819-f002:**
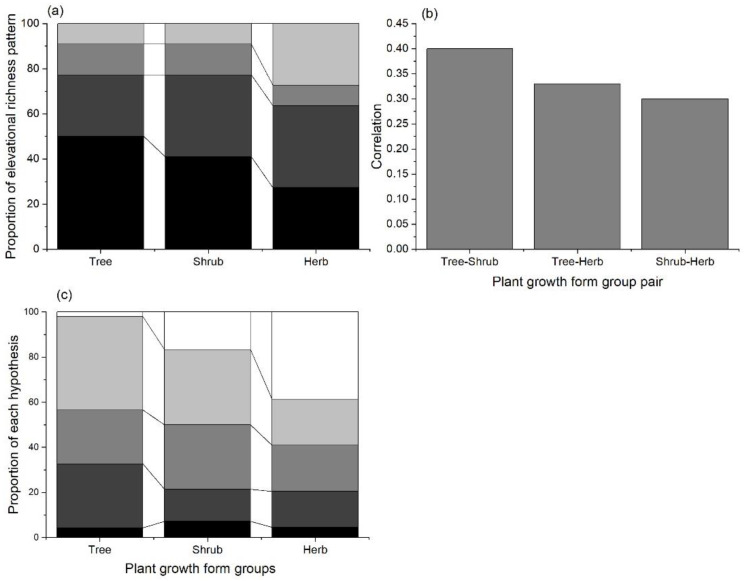
Summary for elevational species richness patterns of plants. (**a**) Proportion of different elevational species richness patterns for each plant growth form group; back: decreasing linearly with elevation; 75% gray: hump-shape with elevation; 50% gray: cubed with elevation; 12% gray: non-significant relationship with elevation. (**b**) Correlation between trees, shrubs, and herbs using the Pearson correlation analysis. The correlation coefficient between the plant groups along the same elevational gradient was calculated. (**c**) The performance of different hypotheses on predicting each elevational species richness patterns, the frequency of occurrence that each hypothesis performed as having the best or second best explanatory power was given; back: area; 75% gray: climatic variability; 50% gray: tolerance; 25% gray: energy; white: unexplainable.

**Figure 3 biology-11-00819-f003:**
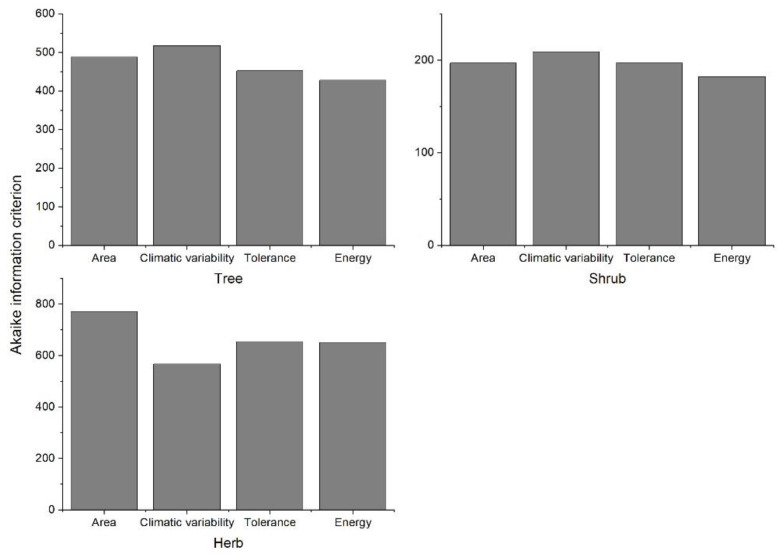
The Akaike information criterion (AIC) for different hypothesis in predicting elevational species richness patterns of each plant growth form group, the average value of AIC of each hypothesis related factor is given.

## Data Availability

All data available after manuscript acceptance.

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
