# Peer review of "The Importance of Energy Theory in Shaping Elevational Species Richness Patterns in Plants"

_biology, 2022, doi:10.3390/biology11060819_

Round 1

Reviewer 1 Report

Dear Authors,

Your manuscript is well written and contributes to a better understanding of the patterns of plant species diversity along an elevational gradient. I have a few suggestions that I think will help improve the paper. First, you used the term taxonomically distinct groups, and in fact they are distinct life forms of plants. Also, the paper does not mention basic characteristics of the research areas. The work is based on published data and lacks own research. Therefore, this study can be considered as a kind of synthesis.

  1. Line 34: Write a comma (,) instead of a dot (.)

, whereas ...

  1. Line44: Maybe it's better to specify and write species richness instead of just richness.
  2. Lines 45, 46 but also the whole text: check if it is written ; or just , after quoting citations.
  3. Line 49: The latitudinal gradient of species richness.
  4. Lines 51-53: maybe to add references for this sentence.
  5. What about the SAR hypothesis? You can cite some studies that look at the relationship between species richness and area as a factor explaining variation in species richness along the elevational gradient.

Species richness along elevation also varies with sampling intensity (Lomolino, 2001): declines or peaks in species richness correspond to declines or peaks in sampling intensity.

Lomolino, M. V. (2001). Elevation gradients of species-density: historical and prospective views. Glob. Ecol. Biogeogr. 10, 3–13. doi: 10.1046/j.1466-822x.2001.00229.x

  1. Delete the dot after the word effects
  2. Line 92-93: Make this sentence to be a question sentence: For example / Are the elevational patterns of species richness for different taxa shaped by different mechanisms?
  3. Did you use your own field data or only published data?

    10. There are numerous studies on the relationship between richness of orchids and altitude. Have you taken them into consideration? I suggest that you mention in the Introduction and Discussion sections the work that looks at patterns of plant diversity along the altitudinal gradient, e.g., in China. There is a need to expand the discussion and compare the results. Some of the papers:

Zhang, S.-B., Chen, W.-Y., Huang, J.-L., Bi, Y.-F., and Yang, X.-F. (2015a). Orchid Species Richness along Elevational and Environmental Gradients in Yunnan, China. PLoS One 10:e0142621. doi: 10.1371/journal.pone.0142621

Zhang, Z., Yan, Y., Tianb, Y., Lib, J., Hea, J.-S., and Tanga, Z. (2015b). Distribution and conservation of orchid species richness in China. Biol. Conserv. 181, 64–72. doi: 10.1016/j.biocon.2014.10.026

Acharya, K. P., Vetaas, O. R., and Birks, H. J. B. (2011). Orchid species richness along Himalayan elevational gradients. J. Biogeogr. 38, 1821–1833. doi: 10.1111/j.1365-2699.2011.02511.x

  1. At what altitude (altitude ranges) are the study areas? I think it would be good to write basic things about study areas.

  1. In the Materials and Methods, it is important to emphasize how you divided the plant groups. You wrote that you worked with different taxonomic groups, and in my opinion you worked with different life forms. Throughout the text, it is necessary to reconsider the use of terms for taxonomically distinct groups and see if it is better to use different life forms as a term.

  1. Table 1. Area - It is necessary to add a reference and a dot after the sentence about the importance of the area.

  1. Line 240: You used a comparison of plant life forms rather than taxonomic structure.

  1. The section Discussion: Try to further compare your data with other studies in China and other parts of the world.

Reviewer 2 Report

In this work the authors conducted a multiple comparative study for trees, shrubs, herbs, and ferns along the same elevation gradient for 12 mountains systems. Particularly, they examined the performance of several hypotheses explaining richness patterns along elevation: energy, tolerance, climatic variability, and spatial area.

The subject is interesting but the presentation for the work needs improvement for what concern the description of the 4 hypotheses both in the introduction, methods and discussion. Language needs revision.

Specific point

61-62: The Energy Hypotesis should be explained more in detail. This is important to furnish the readers an adequate background of the study. Please add e paragraph.

84: Not clear what refers “the same elevation gradient” do you mean elevation range? Of which extent? Please, specify.

89: I missed the other three hypotheses. The authors have to adequately introduce and describe them in the Introduction

114—115: Please rephrase sentence.

120-122. The formulas must be better described: What are constant numbers? What is L? A temperature gradient? described.

169-170/Table1. These hypothesis must be adequately presented in the introduction

183-185. What about the range of elevations you invesigated?

192(Figure2/3) Why didn’t you present a grouped trend for herbs, shrubs and trees? This greatly would help to understand the general pattern of biodiversity. In addition in several works the results are not split according to the life forms

210: Revise sentence (remove second of from “of each of”)

Table 2: Table 2 is not symmetric. Area should be at the same level of Climate var., Tolerance and Energy.

Tabe 2 (Second column): Change the heading since Tree, shrub and herb are not taxa but life forms or growth forms.

Tabe 3 (Second column): Change the heading since Tree, shrub and herb are not taxa but life forms or growth forms.

Table 3. Please describe in the table legend the fourth column “Model remove collinearity”. Not clear

Round 2

Reviewer 1 Report

Dear Authors,

you answered to all my suggestions and all questions correctly, so I think the manuscript can be accepted.

Reviewer 2 Report

Accept